# Gene drive and genetic sex conversion in the global agricultural pest *Ceratitis capitata*

Angela Meccariello [1] ✉, Shibo Hou [1], Serafima Davydova [1], James Daniel Fawcett [1], Alexandra Siddall[2], Philip T. Leftwich [2], Flavia Krsticevic[3], Philippos Aris Papathanos [3] & Nikolai Windbichler [1] ✉

Homing-based gene drives are recently proposed interventions promising the area-wide, species-specific genetic control of harmful insect populations. Here we characterise a first set of gene drives in a tephritid agricultural pest species, the Mediterranean fruit fly *Ceratitis capitata* (medfly). Our results show that the medfly is highly amenable to homing-based gene drive strategies. By targeting the medfly *transformer* gene, we also demonstrate how CRISPR-Cas9 gene drive can be coupled to sex conversion, whereby genetic females are transformed into fertile and harmless XX males. Given this unique malleability of sex determination, we modelled gene drive interventions that couple sex conversion and female sterility and found that such approaches could be effective and tolerant of resistant allele selection in the target population. Our results open the door for developing gene drive strains for the population suppression of the medfly and related tephritid pests by co-targeting female reproduction and shifting the reproductive sex ratio towards males. They demonstrate the untapped potential for gene drives to tackle agricultural pests in an environmentally friendly and economical way.

Homing gene drives were originally proposed 20 years ago[1] are now under development in multiple insect species of medical, agricultural, or ecological importance[2-13]. Suppressive homing gene drives are designed to reduce the population size of harmful pest organisms by, in most cases, targeting genes essential for insect fitness using a homing CRISPR endonuclease expressed in the germline. As a result, the homing construct, strategically placed within the target gene thereby disrupting it, is transmitted at rates exceeding Mendelian inheritance (gene drive). In most insect species, the number and productivity of females determines population size, and often crop damage or disease transmission occurs solely through the activities of female[14]. As such, approaches that interfere with female development or shift the sex ratio towards males are among the most promising strategies being explored[15]. No gene drive has ever been tested in the field, however for medically relevant mosquito species like the malaria mosquito *A. gambiae*, these technologies have reached a stage where

such gene drives can reliably eliminate mosquito populations in the laboratory[16-18]. Consequently, efforts are now being made to establish the technical, social, and regulatory frameworks for field testing and potential deployment.

Tephritid fruit flies, such as the Mediterranean fruit fly (medfly) *Ceratitis capitata*, represent a large group of economically important agricultural pests[19] that are exceedingly well suited for expanding the development of suppressive gene drives beyond mosquitoes. First, they are among the most important agricultural fruit fly pests affecting global food production[20]. Second, many species are non-endemic in affected areas, exacerbating their impact; for example, the medfly in the Americas[21-23]. The current gold-standard for Tephritid population suppression worldwide is biological control, specifically the Sterile Insect Technique (SIT)[24]. Although transgenic approaches have been developed[25], classic SIT, based on genetic sexing strains and pupal irradiation, has seen the most widespread adoption and success for the

[1]Department of Life Sciences, Imperial College London, London SW7 2AZ, UK. [2]School of Biological Sciences, University of East Anglia, Norwich Research Park, Norwich NR4 7TJ, UK. [3]Department of Entomology, Robert H. Smith Faculty of Agriculture, Food and Environment, Hebrew University of Jerusalem, Rehovot, Israel. ✉e-mail: a.meccariello@imperial.ac.uk; n.windbichler@imperial.ac.uk

last 30 years. The success of these SIT programs was built on decades of research and development that can now be used to facilitate the rapid development of gene drives in these species. This includes not only a deep understanding of species-specific traits that could now be leveraged, but also operational know-how on mass-rearing, release strategies and post-release monitoring.

Recently, CRISPR/Cas9 based genome editing has been established in a number of important Tephritid species. In the medfly specifically, an endogenous toolbox for CRISPR genome editing was established, including regulatory elements for germline and somatic Cas9 and sgRNA expression, fluorescent markers and endogenous markers to score gene editing[26]. This progress has coincided with the completion of functional genetic studies mapping the sex-determination pathway, leading to the identification of the Y-chromosome linked male-determining factor *MoY*, that overwrites default female development[27–29]. Interestingly, unlike mosquitoes or *Drosophila*, the pathway is unusually malleable when perturbed. For example, the generation of XX fertile males and XY fertile females is possible[28]. This opens exciting opportunities for genetic control through true genetic sex conversion, an impossibility in many other insects due to the role of the Y chromosome for male fertility, or dosage compensation of X-chromosome linked genes, as for example in the *Drosophila* family[30,31].

Here we sought to test a first set of gene drive constructs in the medfly targeting the known phenotypic marker gene *white-eye* using different regulatory elements to evaluate their performance in the medfly germline. We also sought to target the medfly sex determination pathway to explore the possibility of generating sex conversion gene drives in the medfly and related Tephritid species.

## Results

### Evaluating CRISPR-Cas9 induced homing in the medfly germline

We first established homing-capable gene drive constructs within the well-studied *white-eye* gene[32] (*white-eye*, chromosome 5, GeneID_101458180) to test whether the medfly germlines are generally amenable to homologous recombination-based repair (Fig. 1). Targeting *white-eye* also allows us to utilize existing *white-eye* mutant strains to measure Cas9 paternal effects as well as contrast cutting and homing rates by observing eye pigmentation in the progeny. For this purpose, we placed Cas9 under the transcriptional control of three conserved regulatory elements that have been used extensively in other insect species. These included orthologs of the *nanos*[33] (GeneID_101451248), *vasa* (GeneID_101456741) and *zpg* (GeneID_101449162) genes, from which we extracted both 5' and 3' regulatory regions. *Vasa* has at least five predicted splicing isoforms. To capture all possible 5'UTR variants and the endogenous promoter sequence, we removed 5951 bp of intronic sequence between transcript variant X2 and X3, retaining only the splice junctions while preserving 978 bp of 5'UTR and 1.5 kb of 5' putative promoter sequence. The final gene drive constructs (Fig. S1, Source Data file) also included the *pUb-DsRed* marker gene and an expression cassette for a previously characterised gRNA targeting exon three of the *white-eye* gene[26,34]. All constructs contained approximately 1 kb homology arms flanking the gRNA site, for insertion by homology-based knock-in, directly within the target site of *white-eye*, thereby disrupting it. The plasmids were injected into wild type (Benakeion) embryos and transgenic F1 individuals displaying DsRed fluorescence were used to establish each transgenic strain through individual crosses to Benakeion flies.

To determine the rate of inheritance bias, we first crossed transgenic females or transgenic males to either *white-eye* homozygous mutant (Fig. 1, upper panels) or wild type medflies (Fig. 1, lower panels) and scored progeny for the DsRed marker. For all crosses involving transgenic females, we observed significant super-Mendelian inheritance above 70% (Fig. 1B), from which *nanos* induced the highest mean level of gene drive ($\bar{x}$=85.6%). Activity in the male germline generally

resulted in lower transmission bias, compared to the female germline for each construct. The highest transmission bias in males was driven by the *vasa* promoter, with no observable transmission bias for constructs featuring the *nanos* regulatory regions driving Cas9. For the *zpg* strain, the cross to *white-eye* mutant females ($\bar{x}$=57.5%, $X^2$ = 22.1, p < 0.001, chi-square goodness-of-fit test) revealed significant levels of super-Mendelian transmission but not to wild type females ($\bar{x}$=51.8%, $X^2$ = 0.71, p = 0.3989, chi-square goodness-of-fit test). Overall, the *vasa* construct showed balanced levels of drive in both sexes achieving a level of 78.1% and 70.7% mean transmission in the cross to wild type males and females, respectively. These results indicate that the medfly is generally amenable to homing-based gene drives in both sexes, and that gene drive performance could be further optimised in males by regulating Cas9 expression levels and timing.

### Evaluating CRISPR-Cas9 activity and parental embryonic deposition in the medfly germline

To further correlate homing rates in these crosses with levels of Cas9 activity and to measure rates of parental CRISPR-Cas9 deposition, we analysed the DsRed negative progeny i.e., non-carriers of the drive element (Fig. 1C) for eye colour phenotypes caused by disruption of the *white-eye* gene. Crosses to *white-eye* mutant medflies revealed that for transgenic mothers on average 74.7% (*nanos*), 35.4% (*vasa*) and 49.5% (*zpg*) of chromosomes, that had not participated in gene conversion via homing (i.e DsRed-), carried disrupted target sites. These events are likely the result of non-homologous end joining repair following cleavage in the germline giving rise to white eyed progeny (Fig. 1C, upper panel). Male transgenic crosses yielded mean mutation rates of 5.5% (*nanos*), 36.2% (*vasa*) and 34.8% (*zpg*) indicating that lower Cas9 levels in testes may explain the lower rates of homing observed for *nanos* and *zpg*. Overall, the large fraction of wild type, dark eyed (red eyed) individuals, indicative of unmodified chromosomes, in these experiments suggested that the Cas9 expression levels achievable with these promoters are likely limiting. No cases of somatic mosaicism with regards to eye phenotype were detected in the crosses to the *white-eye* mutants, as expected.

To evaluate parental embryonic deposition and post-zygotic activity of Cas9 we compared eye phenotypes in crosses of transgenic males and females to wild type individuals (Fig. 1C, lower panel). In the cross of transgenic females to wild type males, activity of Cas9 acting on the paternal unmodified copy of *white-eye* is necessary for white eyed progeny to arise. In line with this expectation, we observed reduced levels of mutant progeny and some evidence of somatic mosaicism in the offspring of *nanos* (42.2% white eyed progeny) and *zpg* (7.5% white eyed progeny) drive females (Fig. S2A). The *vasa* construct yielded 87.1% mutant progeny, suggesting very high levels of maternal deposition in this set of experiments. Finally, we observed no white eyed progeny when male transgenics were crossed to wild type females, except again for the *vasa* crosses where sporadic occurrence ($\bar{x}$=0.049%) of white-eyed progeny suggests the possibility of paternal carryover of Cas9. The observed trends were broadly confirmed when we scored the eye phenotype of the DsRed positive fraction of the progeny (Fig. S2B) although this set of experiments allows no distinction between somatic Cas9 expression and Cas9 carryover.

### Generation of a gene drive targeting the medfly transformer gene

We next sought to exploit these insights to target the *C. capitata transformer* gene (*Cctra*, chromosome 6, GeneID_101456163). Since the *vasa* promoter offered the best overall activity in the male and female germlines, we used it to establish a gene drive construct located directly within the *Cctra* locus (*tra* drive), thereby disrupting it. The construct expresses a single gRNA targeting exon one of *tra* (Fig. S1B). Following the establishment of this strain from a G0 founder male we did not observe the occurrence of transgenic females in the progeny,

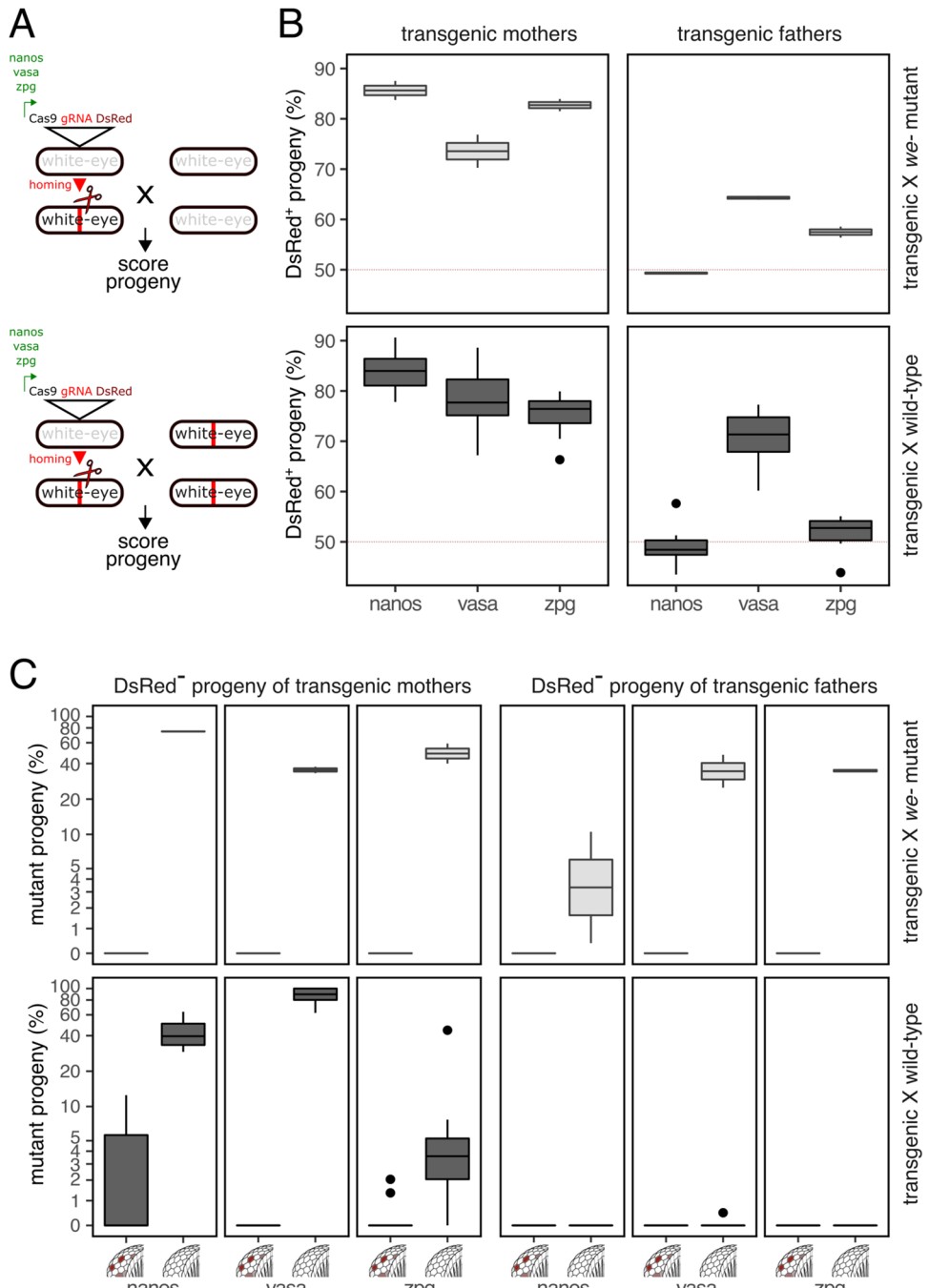

**Fig. 1 | Homing and Cas9 activity targeting *white-eye*. A** Genetic crossing scheme of transgenics crossed to *white-eye* (*w-*) mutant (upper panels) or wild type (lower panels) medflies and schematics of the gene drive constructs tested. **B** Gene drive transmission as the percentage of DsRed positive progeny of female or male hemizygous transgenics. **C** Eye pigmentation phenotypes of the DsRed negative fraction of the progeny of transgenic fathers or mothers. Shown is the percentage of white eyed and mosaic progeny obtained for each construct with red eyed progeny making up the remainder. The total number of individual offspring scored for male and female transgenic crosses were n = 3644/n = 2521 for *nanos*, n = 5115/n = 3622 for *vasa* and n = 3770/n = 3916 for *zpg* constructs respectively. Pooled crosses (10 males x 20 females) of *white-eye* mutant and wild type individuals were performed in 2 and 10 replicates respectively. Boxplots show median values (line), the interquartile range (IQR, box), minima & maxima (whiskers) and outliers beyond 1.5 times the IQR. Source data are provided as a Source Data file.

instead observing males and intersex individuals showing either complete or partial "patchy" expression of the DsRed marker (Fig. S3A). Over multiple generations, we crossed transgenic males carrying the *tra* drive to wild type females (Fig. 2A). The overall rate of transmission was 83.1%, a higher rate of gene drive than observed from within the *white-eye* locus (Fig. 2B). When we analysed the DsRed negative progeny for sexual phenotypes, we found a significant proportion of intersex individuals (20.0%) in addition to males (44.7%) and females (35.1%) (Fig. 2C and Fig. S3B). Strikingly, among the DsRed positive progeny no transgenic females were obtained. The progeny consisted exclusively of intersexes (35.3%) and males (64.6%). Overall, males represented 61.3% of the total progeny. To better understand the observed phenotypes, we carried out molecular analysis of the *tra* gene by Sanger sequencing, as well as karyotyping PCRs of DsRed positive male and intersex individuals utilising sex-chromosome specific PCR primers on genomic DNA. This analysis (Supplementary

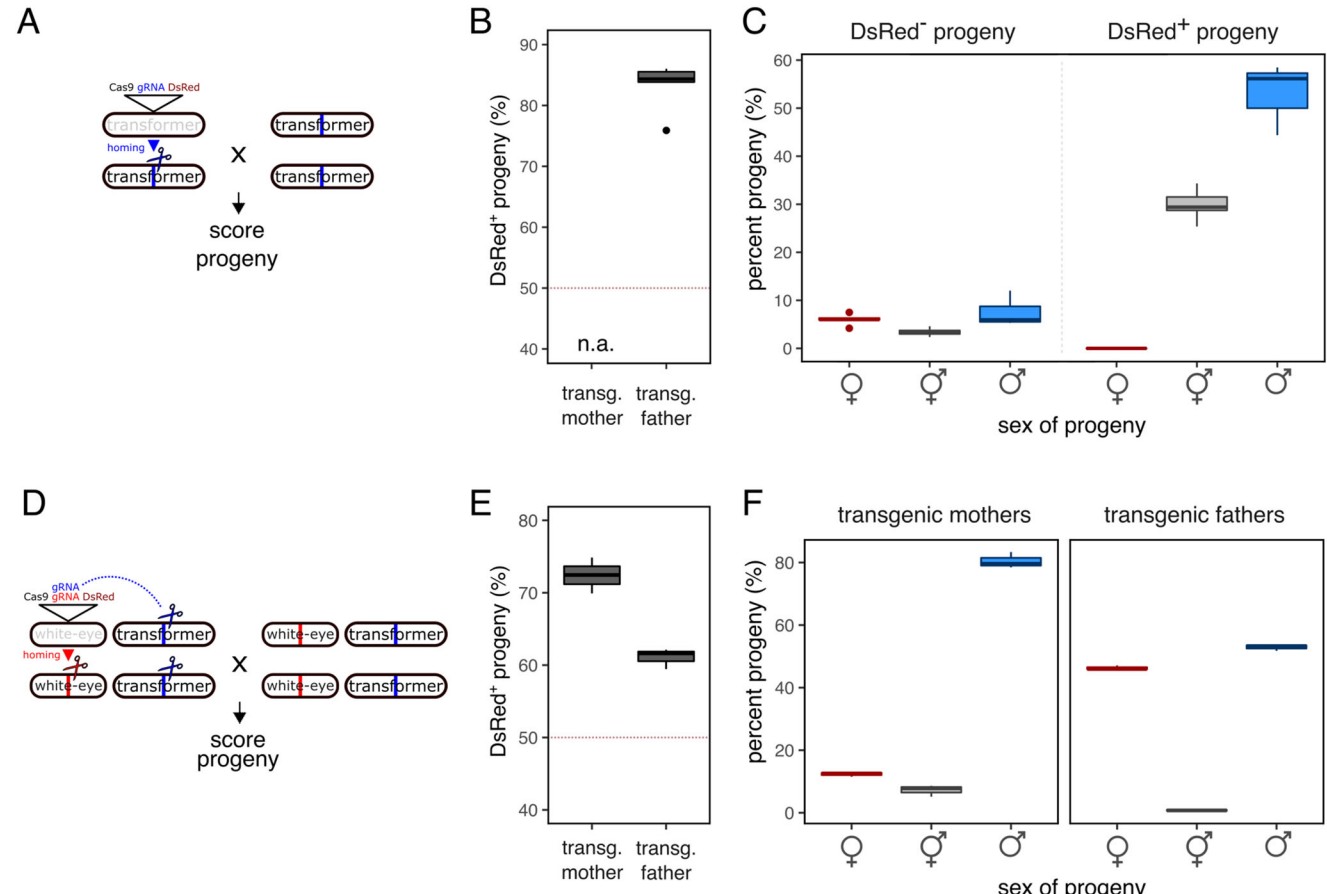

**Fig. 2 | Sex converting gene drive in the medfly. A** Schematic showing the *tra* drive and the crossing scheme used to score the progeny. **B** Transmission of the *tra* drive as the percentage of DsRed positive progeny of male hemizygous transgenics. No female transgenics were obtained. **C** Overall percentage of DsRed positive and negative individuals and their sexual phenotypes in the progeny of *tra* drive males. **D** Schematic showing the *white-eye* + *tra* drive and the crossing scheme used to score the progeny. **E** Transmission of the *white-eye* + *tra* drive as the percentage of DsRed positive progeny of male and female hemizygous transgenics. **F** Sexual phenotypes scored in the progeny of *white-eye* + *tra* drive males or females crossed to the wild type. The total number of individual offspring scored for the *tra* drive was n = 8361 from 5 replicate pooled crosses of 10 *tra* males to 20 females. The total number of individual offspring scored for 3 replicate pooled male and female transgenic crosses (10 males x 20 females) of the *white-eye* + *tra* drive was n = 2173 and n = 3084 respectively. Boxplots show median values (line), the interquartile range (IQR, box), minima & maxima (whiskers) and outliers beyond 1.5 times the IQR. Source data are provided as a Source Data file.

Table 1) revealed that most individuals inheriting the *tra* drive showed, with regards to the amplicon from the maternally inherited chromosome, mixed sequencing traces starting near and centred around the predicted Cas9 cleavage site, and thus were genetic mosaics with regards to the *tra* target site. This suggests that leaky somatic Cas9 expression and somatic cleavage of *tra* resulted in sex conversion of females to XX intersex and XX males, which most likely accounts for the observed sex bias. No clear correlation was observed between the genotypes and sex chromosome karyotypes and the DsRed fluorescence pattern (complete or patchy).

We next characterised the fertility of XX males and intersex individuals arising from crosses of the *tra* drive. By karyotyping 20 transgenic and 10 non-transgenic males that had been crossed to wildtype females individually we identified three transgenic XX male and two non-transgenic XX male individuals. All successfully sired offspring, with the XX non-transgenic males giving rise to all-female progeny (Supplementary Table 2). Interestingly, the 3 XX transgenic males we analyzed (despite passing on the construct via gene drive) did not themselves give rise to XX transgenic sons with intersex individuals making up the bulk of their progeny (Supplementary Table 2). These findings thus corroborate previous studies, suggesting that XX males are viable and fertile[35] but also hint that a full account of the interplay between transgene and *tra* function requires further investigation. By

contrast, none of the crosses of XX intersex individuals yielded progeny (Supplementary Table 3). Despite these limitations of the *tra* drive construct, i.e., postzygotic activity and somatic targeting of *tra*, our transgenic approach demonstrates how gene drive and true sex conversion of females into fertile XX males could be achieved by a single construct in *C. capitata*.

## Generation of a sex conversion gene drive disrupting the maternal provision of transformer

In the medfly, maternal *tra* gene product initiates a regulatory loop necessary for female development[27]. Disrupting *tra* expression in the ovary and inheritance via the female germline would thus lead all progeny defaulting to the male developmental pathway, and it could therefore be a component of an attractive genetic control strategy based on sex conversion. To explore this strategy, we generated another strain where we supplemented the previously characterised gene drive construct located within and targeting *white-eye* (utilising the *vasa* promoter) with an additional gRNA targeting transformer (*white-eye* + *tra* drive) in trans (Fig. 2D). In this strategy the expression of two gRNAs was designed to both bias inheritance of the transgene via homing into the *white-eye* locus and, at the same time, disrupt the *transformer* gene in the female germline by inducing disruptive indels. For this set of experiments, we were able to obtain transgenic

individuals of both sexes, which were then crossed to wild type medflies. We then analysed DsRed fluorescence, as well as sexual and eye colour phenotypes in the resulting progeny.

First, we recorded transmission rates of 72.4% and 61.0% in female and male transgenic crosses, respectively (Fig. 2E), rates of gene drive that were in line with the pilot experiments featuring the *white-eye* drive with a single gRNA. Male transgenics gave rise to a weakly male-biased progeny (52.8%, $X^2 = 9.2$, p = 0.002, chi-square goodness-of-fit test) with the sporadic occurrence of intersex individuals (Fig. 2F). These results were consistent with our expectations given the nature of the *Cctra* function when mothers are wild type. By contrast, we observed a strong male bias in the progeny of transgenic mothers, with males and intersexes representing 80.3% and 7.3% of the total offspring, respectively (Fig. 2F). These results suggest a strong maternal effect on the sex ratio of the progeny of *white-eye + tra* females. With regards to the *white-eye* locus, the analysis of eye colour phenotypes in these crosses also confirmed a strong maternal effect, which led to progeny consisting almost exclusively of white-eyed individuals (Fig. S4). By contrast, only a small fraction of mosaic (6.9%) or white eyed (0.79%) individuals was identified in the progeny of transgenic males, the majority of which were scored as DsRed positive (92%) offspring. Since targeting *tra* in the ovary could result in an unintended reduction in female fitness, we also analysed the number of eggs laid by females (a measure of fecundity) and the number of eggs that hatched among those laid (a measure of fertility) of the *white-eye + tra* drive strain and compared it to all other transgenic strains generated in this study (Fig. S5). We found no significant differences in fertility of either male or female transgenic medflies compared to the wild type control. In summary, these results hint at a hitherto unexplored molecular mechanism, by which the reproductive sex ratio could be shifted towards males, whereby transgenic females would predominantly give rise to transgenic sons.

## Modelling sex conversion gene drives

The *white-eye + tra* drive as currently constituted does not alter female fertility as is the case in conventional suppressive gene drive designs that have been tested e.g. in mosquitoes. However, if it were to be retargeted towards a recessive female fertility gene, it could constitute a powerful new intervention strategy, we term sterilising sex conversion (SSC, Fig. S6). When SSC males are crossed to wild type females, super-Mendelian inheritance of the construct and transmission of a disrupted *tra* allele occurs. However, in SSC females, in addition to the above, the provision of maternal *tra* would be disrupted leading to the generation of male only XY and XX progeny. As the SSC construct spreads, the progressive disruption of *tra* in the population would lead to an extreme male-bias, while the inactivation of the female fertility gene (via homing and indel formation) would ensure that the remaining females are increasingly likely to be sterile. As the population becomes increasingly male biased, functional resistant alleles of the female fertility gene are more likely to be found in a phenotypic male, thereby limiting their relative selective advantage over non-functional alleles. To evaluate this possibility, we modelled this strategy using an agent-based discrete-generation gene drive model and compared it to canonical gene drives directly targeting either female fertility (Fig. 3A, E, I) or directly targeting *transformer* function (Fig. 3B, F, J) which have previously been modelled[30] (Fig. 3). Briefly, we studied the effect of overall Cas9 activity i.e., the rate of cleavage in the germline (and a proportional maternal effect) and the effect of the functional conservation of the target site, i.e., the rate at which non-functional R2 mutations are generated as opposed to R1 resistance mutations that maintain target gene function. The rate of homology directed repair (HDR) vs. non-homologous repair was the third dimension analysed in our model.

We find that the SSC strategy (Fig. 3C, G, K) exhibits elevated tolerance for functional drive resistance. This is because SSC targets

females in two synergistic ways: First, it can convert genetic daughters into fertile sons by cleaving the master female gene *transformer* in the germline, thereby abolishing maternal *tra* function, without which all progeny of SSC females develop as fertile males. The gRNA targeting *tra* is co-expressed from within a gene drive targeting female fertility. Since both gRNAs are always coinherited in the single SSC construct, its presence rarely affects the fertility of the individual harbouring it (homozygous carriers are fertile because they are also always male) and thus the spread of the drive results in rapid population suppression. For the same reason, SSC also sidesteps the issue of maternal deposition of Cas9 known to cause unwanted high rates of non-homologous end joining (NHEJ) in the embryo. Although maternal deposition of Cas9 in the embryos could generate R1 alleles at the female fertility target, these would not immediately be selected for, because the lack of maternal *tra* would cause such embryos to develop into males. SSC thus pre-empts and delays counterselection rendering it more robust against functional resistance at each target gene in our model.

## Discussion

We describe the establishment of homing gene drives in a Tephritid agricultural pest species, the Mediterranean fruit fly *Ceratitis capitata*. Levels of homing recorded in the literature for different classes of target organisms vary dramatically with some species (e.g. *Anopheles* mosquitoes) showing near optimal levels, whereas for others (e.g. rodents) homing has not been shown to always occur at appreciable levels in the germline[13,36]. The levels of gene drive we observed in the medfly using the CRISPR-Cas9 system and employing a set of first generation of germline-specific promoter elements suggest that homing strategies are viable and attractive in this species and could be used to develop effective population suppression tools for this important pest. At the *white-eye* locus, homing in the female germline, was generally higher than in males, and the fraction of uncut chromosomes we observed in both sexes suggests that Cas9 activity levels, rather than a tendency towards non-homologous repair outcomes, was the rate-limiting factor for homing. Only the *vasa* promoter supported substantial levels of drive in both sexes. Improved regulatory elements should therefore be identified and tested as a next step to generate more effective drives.

We also generated a gene drive construct disrupting the *C. capitata transformer* locus. The medfly *tra* gene provides continuous female-specific Tra function in XX embryos and maintains cell-autonomous female identity. Indeed, it has been shown previously that interfering with *Cctra* expression in XX individuals can trigger complete sexual transformation of both the germline and somatic tissues in adult medflies, resulting in a fertile male XX phenotype[27]. The *tra* drive yielded high levels of homing in males and, unexpectedly, a male-biased progeny in crosses to wild type females. Given that our pilot experiments at the *white-eye* locus revealed a lack of evidence for substantial paternal carryover of Cas9, and since we observed at best a modest sex distortion in the DsRed negative fraction of the *tra* drive progeny, a significant male-biased sex ratio as observed for of the DsRed positive progeny was unexpected. The maternal provision of *transformer* and inheritance of a maternal wild type *tra* allele in this cross should have resulted in an unbiased progeny. Our molecular analysis indicated that somatic activity and sex conversion of females accounted for this observation, and this was most likely caused by the leaky expression of Cas9 in somatic tissues. The majority of XX individuals carrying the *tra* drive were found to be infertile intersexes, although a fraction was fully converted into fertile XX males. Since infertile XX intersex individuals represented a sizable fraction of the DsRed positive progeny of *tra* drive males, we concluded that this drive could not exhibit effective population-wide gene drive. Because no transgenic females could be obtained from it, the limitations which curtailed the usefulness of the *tra* drive as currently constituted also

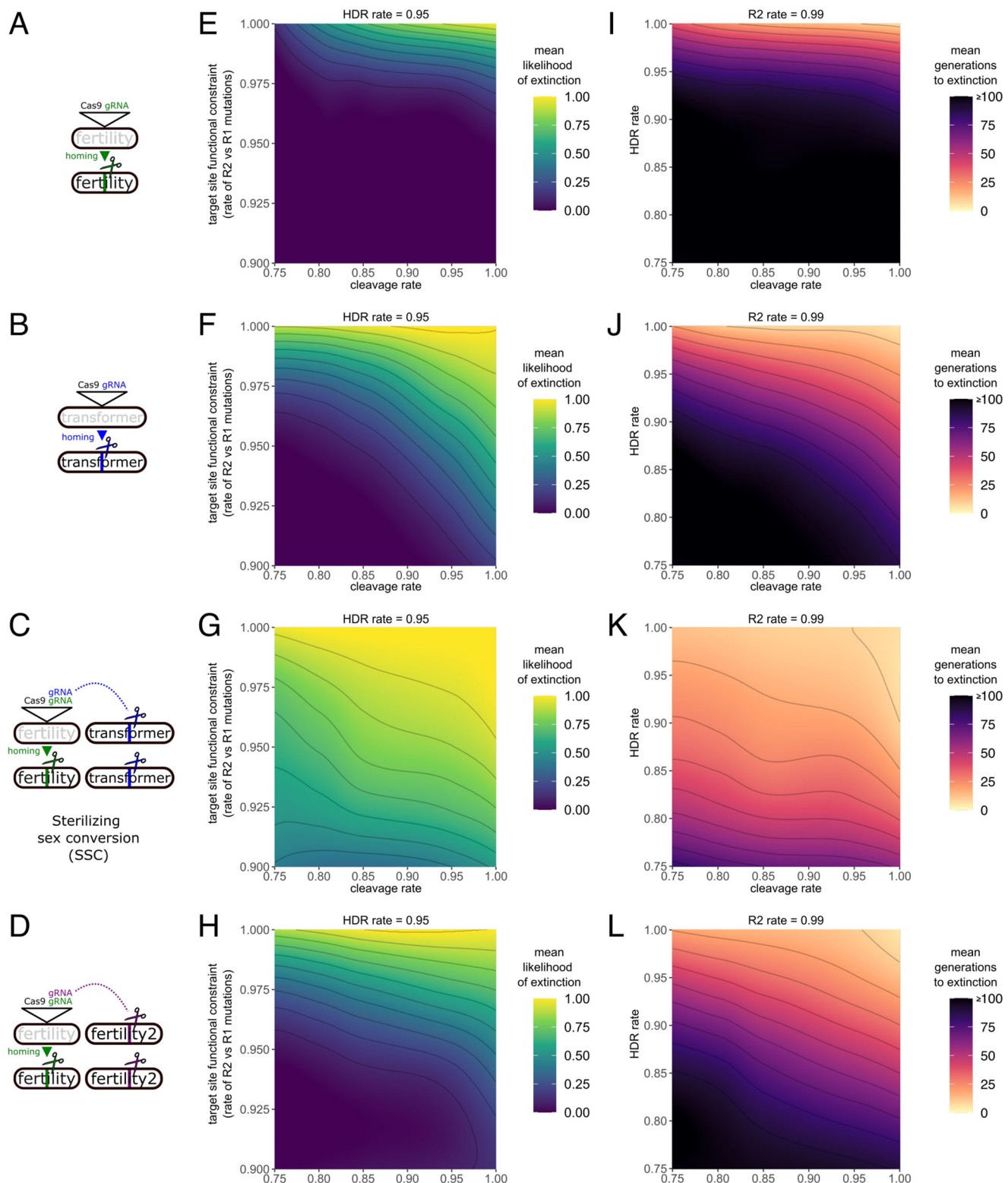

**Fig. 3 | Model of sex conversion gene drive strategies. A** A gene drive construct directly targeting a recessive female fertility gene. **B** A gene drive construct directly targeting the medfly transformer gene. **C** A sterilising sex conversion drive (SSC) homing into a recessive female fertility gene while also targeting the medfly transformer gene in trans. **D** A gene drive construct directly targeting a recessive female fertility gene while targeting a second, separate female fertility gene in trans. Panels **E**–**H** explores the effect of the R2/R1 rate and cleavage rate on the mean likelihood of population extinction at a set HDR rate of 0.95 and a run duration of 20 generations. Panels **I**–**L** explore the effect of cleavage and HDR rates on the mean duration to extinction at a set R2/R1 rate of 0.99/0.01.

prevented us from exploring how the loss of maternal *tra* would affect sex determination.

Targeting the maternal provision of *transformer* is an alternative strategy to effect sex conversion and interferes with the establishment of the autoregulatory loop whereby Tra controls the sex-specific splicing of its own pre-mRNA in females. The *white-eye + tra* gene drive we have generated to test this idea indeed induced a substantial bias towards male progeny in the female, but not in the male transgenic crosses. The results are thus in line with our hypothesis that interfering with ovarian *tra* provision precludes normal female embryonic development in the progeny. However, our experiments also featured strong maternal deposition of Cas9. Further experiments will thus be necessary to evaluate to what degree these two different maternal effects contribute in determining the sex the progeny and to what degree the loss of maternal *tra* results in male development irrespective of the genotype or karyotype of the developing embryo.

In Diptera evidence shows that in the families of Tephritidae (*Ceratitis*, *Anastrepha* and *Bactrocera*)[27,37–39], Muscidae (*Musca*)[40] and Calliphoridae (*Lucilia*)[41] maternal *transformer* (Tra$_{mat}$) and *transformer*2 (Tra2$_{mat}$) gene products initiate a positive feedback regulation of *tra* and enable female-specific splicing of the downstream genes *doublesex* and *fruitless*, unless a male determining factor (e.g. *MoY*, *Mdmd*)[28,42] is present. Furthermore, it has been shown that Tra$_{mat}$ and Tra2$_{mat}$ accumulate in the nurse cells and the growing oocyte but not in the somatic parts of the ovary[43]. Since the *vasa* promoter too is predicted to be active in both of these cell types, it is likely that in our system the level of Tra$_{mat}$ provided to the progeny is the compound result of cleavage and repair events in the 15 nurse cells and the oocyte. This dynamic process should also be explored in further experiments since it is analogous to lethal/sterile mosaicism resulting from maternally deposited Cas9/gRNA, a process that can act dominantly if it occurs in a sufficient number of cells[9,44].

While a number of engineered sex ratio distortion systems and systems that interfere with sex determination have been previously described[26,45–47], our findings demonstrated the possibility of a true sex conversion gene drive that converts females into fertile XX males on the population level. For a gene drive homing directly into *tra* this would however require a tighter control of Cas9 expression at this locus than is currently the case in our *tra* drive strain, for example by making use of insulator elements. At the *white-eye* locus we did not find similar evidence for leaky somatic expression of Cas9, highlighting that ectopic Cas9 expression and its impact likely depend on the genomic context, the target gene with its spatiotemporal pattern of expression.

Our modelling explored coupling gene drive targeting female fertility with genetic sex conversion, a strategy we termed sterilising sex conversion (SSC), as a more attractive option. We show that, compared to a conventional female fertility drive, SSC is faster acting and also more tolerant towards the rates of CRISPR activity, homologous repair and the functional target site conservation achievable. Given the tools we have described in this manuscript, assembling such a system in the medfly should be possible in the near future.

Over the last 40 years there has been considerable progress in the development and integrated application of SIT against the medfly, as reflected by ongoing operational programs for eradication, prevention, and suppression as well as a dramatic increase in sterile fruit fly production capacity on multiple continents. There is however considerable scope for improving the efficiency and economics of medfly control and to develop control strategies for other Tephritid flies, for many of which currently few or no control options exist. As the first demonstration of effecting true and efficient genetic sex conversion in the female germline and the demonstration of gene drive in the medfly, our findings open the door to several alternative control strategies, as well as genetic sexing mechanisms that could be combined with more classical approaches such as the sterile insect technique.

## Methods

### *Ceratitis capitata* rearing and maintenance

Wild type Benakeion and transgenic *C. capitata* strains were bred as described previously[26], specifically under 12:12 hour light:dark cycle, 65% relative humidity and 26 °C. All transgenic strains were established via individual crosses of male transgenic individuals with wild type Benakeion virgin females to confirm the nature of each insertion event.

### Generation of constructs

For sequence analysis we used the EGII-3.2.1 (GCA_905071925.1) medfly assembly. The annotated transformation vectors are provided in Source Data file. Briefly, we used the gRNA (gRNA_white) targeting the *white-eye* gene, as previously described[34] to knock-in into the third exon of the gene (GeneID_101458180). All constructs were generated through a two-step Gibson assembly using the primers indicated (Supplementary Table 4). First, three fragments were cloned into the pUK21 backbone (Addgene #49787) which was digested using XhoI/XbaI to generate the intermediate plasmid pHA-white-DsRed. These included the left and right homology arms, 976 and 998 bp in size respectively, separated by a fragment amplified from piggyBac Cas9.w plasmid[26] containing an AscI restriction site, CcU6-gRNA_white and pUb-DsRed. pHA-white-DsRed was sequentially linearised with AscI for the second assembly in which a cassette was inserted containing Cas9, similarly amplified from the piggyBac Cas9.w plasmid. Three different endogenous germline promoters, *CcNanos*, *CcVasa*, or *CcZpg*, were inserted upstream of Cas9, and their respective endogenous terminators were inserted downstream. The pVasa-w+tra_drive plasmid with an additional gRNA was assembled via linearisation of pVasa-w with FseI and the insertion of the CcU6-gRNA_tra fragment thereafter. This gRNA (gRNA_tra) (GTTGTTATTAAACGTAGATTTGG) was selected using CHOPCHOP v3[48] and targets the first female-specific exon of the *transformer* gene (*Cctra*, GeneID_101456163). Plasmid pGdtra, was similarly constructed using the pUK21 backbone linearised with XbaI/XhoI. First, the left and right homology arms were inserted, 937 and 907 bp in size respectively, separated by the EcoRV restriction site. The resulting plasmid was linearised with EcoRV and fragments amplified from the pVasa-w+tra_drive plasmid were inserted. These contained CcVasa-Cas9, CcU6-gRNA_tra and pUb-DsRed.

### Generation and establishment of transgenic strains

Microinjections into wild-type Benakeion strain embryos of the donor plasmids pNanos-w, pVasa-w, and pZpg-w (250 ng/ml) were performed to generate the 3 *white-eye* drive strains. To generate the *tra* drive and *white-eye + tra* drive strains plasmids pGdtra and pVasa-w+tra_drive were used respectively (250 ng/ml). The injection master mix also included preassembled RNP complex of Cas9 protein (200 ng/ml) (PNA Bio)[34], and either gRNA_white (100 ng/ml) or gRNA_tra (100 ng/ml) obtained from Synthego. Injected G$_0$ adults were reciprocally crossed in pools to wild type virgin flies. The resulting G$_1$ progeny was screened for DsRed at the adult stage, and fluorescent individuals were individually crossed to wild type virgin flies of the opposite sex. To confirm knock-in of the homing constructs into *white eye* and *transformer* target regions, gDNA from single flies was extracted using an adapted protocol from Holmes & Bonner[34,49] using a phenol-chloroform method of DNA isolation. PCRs were then carried out with primers binding to the homing construct and genomic sequence outside the homology arms (Supplementary Table 4).

### Genotyping and sex chromosome karyotyping

Template gDNA from single flies was used for genotyping, karyotyping, gRNA-white target and gRNA-tra target PCRs (Supplementary Table 4). The genotyping to identify the drive allele and the wild type allele was performed with a multiplex PCR. For the *white-eye* drive line, the following primers were used; F-indel-w binding upstream to the cleavage site of *white-eye* gene and For_DsRed binding in *white-eye*

*drive cassette* and reverse primer binding in the genome Rev_Genome-w. While for the tra drive line the following primers were used: For_Pub binding in *tra drive cassette*, F-indel-tra binding upstream to the cleavage site of *tra* gene and a reverse primer binding in the genome of the tra gene Rev_tra-screen. To detect indels at both *we* and tra gene cleavage sites the following primers were used respectively: F-indel-w, R-indel-w; F-indel-tra and Rev-indel-tra. The gRNA target amplicons were purified, and Sanger sequenced. To perform sex chromosome karyotyping of XX and XY individuals PCRs were conducted using RedTaq DNA Polymerase 2X Master Mix (VWR Life Science) with the following published primers: *CcYF/CcYR*[50].

### Fecundity and fertility assays
Egg-laying rates and egg-hatching rates were assessed by crossing 10 transgenic males or intersexes with 20 wild-type Benakeion virgin females in triplicates, alongside three control crosses of 10 wild-type males and 20 wilt-type females. Upon maturation, eggs laid during a 5-hour period were collected on a black filter paper, images and counted immediately using ImageJ. After a further 4 days, unhatched eggs were counted again to determine the egg hatching rates. Statistical analysis was performed using an ANOVA (Dunnett's test).

### Homing assays
Male and female transgenics were reciprocally mated with wild type Benakeion flies in standard crosses of 10 males and 20 females with the exception of the *tra* drive, where only male transgenics were available. The progeny of the *white-eye* drive lines (*nanos, vasa* & *zpg* promoters) were screened for eye colour (red/mosaic/white) and fluorescence phenotype (DsRed+/DsRed-) for 10 consecutive generations. The progeny for the *white-eye + tra* drive crosses were additionally screened for phenotypic sex characteristics (male/female/intersex), while the *tra* drive cross progeny were screened for fluorescence phenotype and phenotypic sex for three and five consecutive generations respectively. Eggs were collected at two timepoints, with an interval of 2 to 3 days, and reared to adulthood. Adult flies were screened using the MVX-ZB10 Olympus with an RFP filter (excitation filter 530–560 nm, dichroic beam splitter 570 nm, barrier filter 585–670 nm). Statistical analysis was performed using the chi-square goodness-of-fit test.

### Gene drive model
We employed the SMS agent-based gene drive model[51], where we considered populations of 1000 individuals with discrete generations. All gene drive designs were investigated at a starting allele frequency of 12.5% with populations consisting initially of 500 wild-type females, 250 wild-type males, and 250 hemizygous drive males. The model considers fertility effects and target site resistance at each locus independently. Parameter sweeps were performed for the rate of Cas9 germline activity, the rate at which R1 (functional) versus R2 (non-functional) resistance alleles are formed at each target locus, as well as the HDR rate i.e. the rate at which cleavage leads to homologous (homing of the construct or of an R allele present at the other chromosome) versus non-homologous (R allele formation) repair. When Cas9 germline activity levels were varied we considered maternal Cas9 activity to vary proportionally. Each model is based on 360 simulations covering the parameter space indicated. The continuous rate of extinction was then calculated using a local polynomial regression model.

### Reporting summary
Further information on research design is available in the Nature Portfolio Reporting Summary linked to this article.

## Data availability
All data needed to evaluate the conclusions in the paper are present in the paper. Input data and scripts to generate the Fig. panels presented in the paper available at https://github.com/genome-traffic/medfly_genedrive_paper (https://zenodo.org/records/10283780). We utilized the Ccap2.1 reference genome assembly, the EGII-3.2.1 genome assembly as well as the Ccap2.1 reference annotation available via Genbank accessions GCA_000347755.4, GCA_905071925.1 and GCF_000347755.3 respectively. Source data are provided with this paper.

## Code availability
The SMS model code and parameters are available at https://github.com/genome-traffic/SuperMendelianSandbox (https://zenodo.org/records/10283751) and R scripts for visualisation are available at https://github.com/genome-traffic/gRandTheftAutosome (https://zenodo.org/records/10283730).

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

## Acknowledgements

This study was funded by the Biotechnology and Biological Sciences Research Council (BBSRC) under research grant BB/W00304X/1 to N.W. and A.M., Imperial College Research Fellowship 2020 to A.M. and the Israel Binational Agricultural Research and Development Fund (BARD Research grant no. IS-5180-19) to PAP. A.S. is supported by a UKRI BBSRC Norwich Research Park Biosciences Doctoral Training Partnership (Grant No. BB/M011216/1 awarded to PTL and Professor Tracey Chapman). We thank Ernst Wimmer and Sebald A.N.R. Verkuijl for their advice and comments.

## Author contributions

A.M. and N.W. conceived the project. A.M. designed and performed the experimental work. A.M. designed constructs. A.M. and F.K. analyzed all the sequences. A.M. and J.D.F. performed the cloning. A.S. and P.T.L. isolated and cloned the vasa promoter. A.M., S.H., and S.D. maintained the strains and performed the crosses and molecular validation experiments. A.M. and N.W. analysed the data. N.W. generated the model. N.W. prepared the Fig.s. N.W. prepared the original draft of the paper. N.W., P.A.P., and A.M. edited and reviewed the paper. All authors read and approved the final manuscript.

## Competing interests

The authors declare no competing interests.
