## [Peer Review File · Nature Communications]

Reviewers' Comments:

Reviewer #1:

Remarks to the Author:

The manuscript by Meccariello et al. describes the first synthetic gene drives of *Ceratitis capitata*, the 'medfly'. Homing gene drives targeting the white-eyed gene and the key sex-determining pathway gene, transformer, are created and analyzed and the progress suggests that gene drives in this species that is already controlled using genetic technique hold a lot of promise. However, it is the fact that fertile XX males can be created in this species that makes this study special because it motivates the exploration of novel "sterilizing sex conversion gene drives" which would work by targeting the drive to a female fertility gene.

There is a lot going on with this paper and my broad comment is that more time needs to be given explaining the more advanced ideas in a more orderly and careful way. It is typical for a gene drive paper to start with some engineering and to end in some modelling that advances upon the engineering experiments that preceded it. Here the engineering was straightforward enough and interesting; eg vasa is a better promoter than nos-cas9 and zpg, the drive is stronger in females than males, and that appears to be attributable to cas9 cutting rather than homing, and there is an open question about whether somatic cas9 or cas9 carryover explains some of the patterns. However, I found the elaborations, including the SSC, confusing.

(1) Therefore my major request is to improve expression with a more patient unfolding of logic. Perhaps an explanatory figure (with the hypothetical design with the female sterile and showing the fate and the resistance of concern) would assist?

The homing drive targeting tra, prevents the female specifying tra protein being produced and so the flies develop as males even if they are chromosomally XX. In retrospect, one component of my confusion started with 'All [XX males] successfully sired offspring, with XX non-transgenic males giving rise to all-female progeny'. At this stage I was wondering what the ultimate skew in sex ratio will be given that the drive creates XX males but they only have daughters. I missed the implied opposite to that sentence, that transgenic XX males still have a drive, and so will produce males. Is that correct? Regardless all this is needs to be teased apart more carefully. Also you can point out that tra knockouts (resulting from tra cutting but not homing) are recessive (rather than on line 211 where the qualification that 'when mothers are wild types' confuses).

(2) Are 'white-eye' and 'transformer' autosomal in medfly?

(3) Line 156 you introduce the term 'in-locus' homing drive. This introduction is out of place isn't it? The white-eyed drive is in-locus and you didn't feel the need to define it as such. The reason you want to use the term is to contrast it with the next drive (line 194 etc) – the remote site drive – but as you haven't introduced that yet the extra terminology is a bit baffling.

(4) There is also an order-of-ideas problem with the paragraphs associated with the initial white-eye drive. Why cross the drive to white eye and to wildtype (lines 104 to 105)? What is the motivation of this design? Are you interested in the effect of genetic background? No, it is to do with cutting versus homing, and that has more to do with the subsequent experiment (lines 123 and on).

(5) Line 172. What exactly is meant by karyotyping? It is not adequately described in the results or the methods and I suspect that the usage is lab jargon for a molecular assay rather than microscopic visualization of chromosome complement but I am not sure.

Minor changes that may improve:

L 168. It is indeed striking that no dsRED females were obtained. Presumably this means that among the daughters there were no gene drive heterozygotes (because hets will be female) and therefore that the vasa-cas9 must be active between syngamy of sperm and egg and when sexual differentiation takes place (as expected given the post-zygotic activity described in lines 138-151). So this suggests that there is vasa-cas9 somatic expression (line 297). So is it worth explicitly

describing the characteristics of a better promoter rather than just referring rather vaguely to the "limitations" – l188 and l302.

Why is nanos expression so divergent between species?

Figure S1B – perhaps illustrate splice patterns (eg female above line and male below) and stop codons in male exons.

L76: I wonder whether the statement on l76 about dosage compensation could be clarified. Do medflies exhibit dosage compensation but it is somehow accounted for in XX males or XY females or do they not have dosage compensation?

The details on the vasa promoter and what was used (lines 93-96) are cryptic. A figure or sequence data in the supp material would be better so that the exact details are unambiguous (although they are not necessary in the first para of the results).

Line 100: "knock-in" rather than "knockin".

L103: backcrossing? If it is a gene drive why is a backcross necessary – presumably you mean to the Benakion parent? Why not simply say 'crossed to Benakion'? I presume you don't inter-cross the F1 because you want to keep inserts independent?

L159: specify B in FigureS1B

I am wondering whether you should refer to 'white+tra' construct of line 199 and elsewhere as 'white-eye+tra' or 'we-tra' because elsewhere you take care to refer to white-eye rather than white. Is white-eye the ortholog of *D.melanogaster* white?

Reviewer #2:

Remarks to the Author:

This paper aims to characterize the first CRISPR based gene drive system in the Medfly. Overall I found the paper well written and only had a few suggestions:

Minor Comments

page 1 line 21-23 - Can authors briefly describe the two different mechanisms by which a CRISPR gene drive induces sex conversion? Otherwise, it is unnecessary unclear.

page 2 line 53 - Replace the semicolon with the full stop.

page 2 line 57 - SIT is biological control. It is not referred to as genetic control.

page 2 line 61 - Replace "fertilize" to "facilitate".

page 2 line 69 - I suggest removing "efficiency markers" and leaving only "and endogenous markers to score gene editing[26]".

page 4 line 144-146 - Remove "the trend did not hold for the vasa construct" In fact, the vasa construct shows the strongest trend of the higher rate of white knockout from the transgenic females.

page 4 line 168-170 - it is not clear if the sentence describes the entire progeny / only DsRed+ progeny / only DsRed- progeny? The numbers do not add up in this sentence. Fig 2C indicates that ~7% of the entire progeny was DsRed- females. Perhaps, it would be good to mention the total DsRed+ progeny % vs that of DsRed-.

page 5 line 210-211 - "recessive nature of the Cctra function" may not be the best description of the process. It seems tra maternal deposition and the tra+ allele of the female wild-type parent complements the tra- allele of the transgenic male.

page 8 line 325-327 - the first reference describing the lethal mosaicism from maternally deposited Cas9/gRNA needs to be corrected: [https://www.nature.com/articles/s41467-018-07964-](https://www.nature.com/articles/s41467-018-07964-7)

7

Major comments.

Fig 1B - It is highly unusual to see the figures with two replicates in the main text. I suggest

conducting additional experiments to obtain the third replicate for Fig 1B. Would it be possible to add one additional replicate?

Authors also need to describe what box plot and whiskers represent in the legend for every figure. The sequence maps of the constructs developed in the study need to be well annotated with major features clearly labeled before the final submission.

REVIEWER COMMENTS

Reviewer #1 (Remarks to the Author):

The manuscript by Meccariello et al. describes the first synthetic gene drives of *Ceratitis capitata*, the 'medfly'. Homing gene drives targeting the white-eyed gene and the key sex-determining pathway gene, transformer, are created and analyzed and the progress suggests that gene drives in this species that is already controlled using genetic technique hold a lot of promise. However, it is the fact that fertile XX males can be created in this species that makes this study special because it motivates the exploration of novel "sterilizing sex conversion gene drives" which would work by targeting the drive to a female fertility gene.

There is a lot going on with this paper and my broad comment is that more time needs to be given explaining the more advanced ideas in a more orderly and careful way. It is typical for a gene drive paper to start with some engineering and to end in some modelling that advances upon the engineering experiments that preceded it. Here the engineering was straightforward enough and interesting; eg *vasa* is a better promoter than *nos-cas9* and *zpg*, the drive is stronger in females than males, and that appears to be attributable to *cas9* cutting rather than homing, and there is an open question about whether somatic *cas9* or *cas9* carryover explains some of the patterns.

We thank the reviewer for their comments.

However, I found the elaborations, including the SSC, confusing. (1) Therefore my major request is to improve expression with a more patient unfolding of logic. Perhaps an explanatory figure (with the hypothetical design with the female sterile and showing the fate and the resistance of concern) would assist?

We took up the suggestion of the reviewer and added Figure S6 which hopefully illustrates the SSC strategy better. In the legend we provide an independent description of the SSC genetics supplementing the explanation in the main text which we have also tried to improve.

The homing drive targeting *tra*, prevents the female specifying *tra* protein being produced and so the flies develop as males even if they are chromosomally XX. In retrospect, one component of my confusion started with 'All [XX males] successfully sired offspring, with XX non-transgenic males giving rise to all-female progeny'. At this stage I was wondering what the ultimate skew in sex ratio will be given that the drive creates XX males but they only have daughters. I missed the implied opposite to that sentence, that transgenic XX males still have a drive, and so will produce males. Is that correct? Regardless all this needs to be teased apart more carefully.

*In the case of the *tra* drive we have the phenomenon of somatic activity as described. Because of this the *tra* drive can't be used for control on a population level because, as discussed, a large fraction of the progeny consists of sterile intersexes. Therefore we have not further teased apart questions about the ultimate skew and possible performance for this specific construct.*

The reviewer is correct in that there is drive in XX male transgenics (we now made this clearer in Supplementary Table 2). There are also a number of observations that suggest

that there is more to the story and could warrant a follow up story on tra function. First, the ~20% intersex individuals in the DsRed negative progeny of tra drive males is higher than what we would have expected to find given the low levels of paternal Cas9 deposition observed for the white-eye locus. Second, the transgenic progeny of the 3 XX transgenic male individuals we looked at consisted exclusively of intersex individuals. In other words, the 3 transgenic XX males despite being fertile did not themselves produce transgenic XX males. The non-transgenic progeny of those same 3 XX transgenic males featured females and no males (as expected) but the majority were again intersex individuals. This points to either high levels of paternal deposition from XX transgenic males carrying the tra drive or possibly a dominant effect of certain mutant tra alleles. We now also added some of this information to the text.

Also you can point out that tra knockouts (resulting from tra cutting but not homing) are recessive (rather than on line 211 where the qualification that ‘when mothers are wild types’ confuses).

This point, also raised by the other reviewer, was clarified.

(2) Are ‘white-eye’ and ‘transformer’ autosomal in medfly?

Yes, both genes are autosomal. The white eye gene maps to chromosome 5 and the transformer gene maps to chromosome 6. This information was also added to the manuscript.

(3) Line 156 you introduce the term ‘in-locus’ homing drive. This introduction is out of place isn’t it? The white-eyed drive is in-locus and you didn’t feel the need to define it as such. The reason you want to use the term is to contrast it with the next drive (line 194 etc) – the remote site drive – but as you haven’t introduced that yet the extra terminology is a bit baffling.

We removed this term and improved the description of the contrasting design in the text.

(4) There is also an order-of-ideas problem with the paragraphs associated with the initial white-eye drive. Why cross the drive to white eye and to wildtype (lines 104 to 105)? What is the motivation of this design? Are you interested in the effect of genetic background? No, it is to do with cutting versus homing, and that has more to do with the subsequent experiment (lines 123 and on).

We now provide more explanation upfront why these crosses were also performed.

(5) Line 172. What exactly is meant by karyotyping? It is not adequately described in the results or the methods and I suspect that the usage is lab jargon for a molecular assay rather than microscopic visualization of chromosome complement but I am not sure.

We now specified that we performed sex chromosome karyotyping using a diagnostic PCR with previously described primers (Gabrieli, P. et al. 2010, BMC Dev. Biol.). Briefly, the PCR

allows us to discriminate between XX and XY individuals and hence can also distinguish between XX and XY males. We clarified this point in the results and in the methods sections.

Minor changes that may improve:

L 168. It is indeed striking that no dsRED females were obtained. Presumably this means that among the daughters there were no gene drive heterozygotes (because hets will be female) and therefore that the vasa-cas9 must be active between syngamy of sperm and egg and when sexual differentiation takes place (as expected given the post-zygotic activity described in lines 138-151). So this suggests that there is vasa-cas9 somatic expression (line 297). So is it worth explicitly describing the characteristics of a better promoter rather than just referring rather vaguely to the “limitations” – I188 and I302.

We specified the limitation directly in the text as suggested by the reviewer.

Why is nanos expression so divergent between species?

This is not necessarily the case. Originally nanos was considered to be expressed in mosquito females only and weakly in Drosophila males. Some gene drive experiments show reduced homing rates in males versus females but generally nanos is now considered to be active in both sexes in mosquitoes. RNAseq data in Drosophila paints a similar picture to what our RNAseq data (unpublished) shows in the medfly, namely strong expression in the ovary (median TPM 150) and lower expression in the males (median TPM of ~4). We did not see Cas9 activity in medfly males using a nanos promoter fragment, however we also don't know whether we have captured all necessary enhancer elements for male and female expression in our synthetic promoter construct.

Figure S1B – perhaps illustrate splice patterns (eg female above line and male below) and stop codons in male exons.

Sex specific splicing patterns and stop codons are now indicated on the figure.

L76: I wonder whether the statement on I76 about dosage compensation could be clarified. Do medflies exhibit dosage compensation but it is somehow accounted for in XX males or XY females or do they not have dosage compensation?

This is a good question. Unfortunately not much is currently known about dosage compensation in the medfly. Having said that, our statement regarding dosage compensation is specifically about other insect species, in which dosage compensation is both known, for example in Drosophila or Anopheles species, and would interfere with a strategy we successfully developed here in the medfly. We know of no formal, published analysis on the question of dosage compensation in the medfly, especially since the X-chromosome assembly remains unannotated.

The fact that XX males are viable and fertile suggests either that if dosage compensation occurs in the medfly (to equalize X-chromosome expression between males and females like Drosophila) then (1) it does not interact with sex determination genes upstream of tra and (2) it is likely not mis-activated in XX males. Alternatively, our results could suggest that there is no dosage compensation in the medfly perhaps even because the medfly X-chromosome

hosts few, essential genes. Since this is mainly speculation at this point, we have opted not to elaborate further on this in our manuscript, but we appreciate this reviewer's curiosity.

The details on the vasa promoter and what was used (lines 93-96) are cryptic. A figure or sequence data in the supp material would be better so that the exact details are unambiguous (although they are not necessary in the first para of the results).

We have provided the annotated sequence file of the transformation vectors. The sentence was merely to illustrate the thinking behind the choice to assemble a synthetic promoter element.

Line 100: “knock-in” rather than “knockin”.

We corrected this in the text.

L103: backcrossing? If it is a gene drive why is a backcross necessary – presumably you mean to the Benakion parent? Why not simply say ‘crossed to Benakion’? I presume you don’t inter-cross the F1 because you want to keep inserts independent?

The reviewer is correct. We do want to analyze insertion events separately, hence we single cross DsRed positive F1 flies to Benakeion wild-type individuals. We clarified this in the methods section.

L159: specify B in FigureS1B

We corrected this in the text.

I am wondering whether you should refer to ‘white+tra’ construct of line 199 and elsewhere as ‘white-eye+tra’ or ‘we-tra’ because elsewhere you take care to refer to white-eye rather than white. Is white-eye the ortholog of D.melanogaster white?

We took up this suggestion and tried to use white-eye more consistently in the manuscript. The reviewer is correct, the white-eye gene in the medfly is the ortholog of the D. melanogaster white gene.

Reviewer #2 (Remarks to the Author):

This paper aims to characterize the first CRISPR based gene drive system in the Medfly. Overall I found the paper well written and only had a few suggestions:

We thank the reviewer for their positive comments.

Minor Comments

page 1 line 21-23 - Can authors briefly describe the two different mechanisms by which a CRISPR gene drive induces sex conversion? Otherwise, it is unnecessary unclear.

Fair point that this may be unclear. Since expanding the explanation in the abstract may take too much space we now state “demonstrate how CRISPR-Cas9 gene drive can be coupled to sex conversion”. in the abstract.

page 2 line 53 - Replace the semicolon with the full stop.

We corrected this in the text.

page 2 line 57 - SIT is biological control. It is not referred to as genetic control.

Opinions may differ here. But we now refer to SIT as biological control in the text.

page 2 line 61 - Replace “fertilize” to “facilitate”.

We corrected this in the text.

page 2 line 69 - I suggest removing “efficiency markers” and leaving only “and endogenous markers to score gene editing[26]”.

We corrected this in the text.

page 4 line 144-146 - Remove “the trend did not hold for the vasa construct” In fact, the vasa construct shows the strongest trend of the higher rate of white knockout from the transgenic females.

The comparison was not to the other promoters, but it appears this was not clear. We followed the reviewers suggestion and removed this passage.

page 4 line 168-170 - it is not clear if the sentence describes the entire progeny / only DsRed+ progeny / only DsRed- progeny? The numbers do not add up in this sentence.

This is well spotted, the sentence was indeed incorrect. We split this sentence in two and made it clearer.

Fig 2C indicates that ~7% of the entire progeny was DsRed- females. Perhaps, it would be good to mention the total DsRed+ progeny % vs that of DsRed-.

This is identical with the transmission rate i.e. the percent of progeny harboring the gene drive so basically what figure 2B describes. In the text we mention that it is 83.1% on average.

page 5 line 210-211 - “recessive nature of the Cctra function” may not be the best description of the process. It seems tra maternal deposition and the tra+ allele of the female wild-type parent complements the tra- allele of the transgenic male.

We simplified the sentence and removed “recessive”.

page 8 line 325-327 - the first reference describing the lethal mosaicism from maternally deposited Cas9/gRNA needs to be corrected:
<https://www.nature.com/articles/s41467-018-07964-7>

The reference was added as the first elaboration of this concept.

Major comments.

Fig 1B - It is highly unusual to see the figures with two replicates in the main text. I suggest conducting additional experiments to obtain the third replicate for Fig 1B. Would it be possible to add one additional replicate?

It is important to state that gene drive occurs in the germline of transhemizygous individuals and doesn't depend on the partner they are eventually mated to. So in effect we don't have 2 replicates but 12 replicates for the gene drive experiment.

The reviewer is correct that paternal effects, where the mating partner indeed matters, are looked at in 10 and 2 replicates separately. Given that we still analyzed 1566 individual progeny we thought that our data is strong enough to support the moderate claims we make for that specific sub-experiment.

However, if the reviewer believes that another replicate is necessary we believe we can generate the additional data by the end of November.

Authors also need to describe what box plot and whiskers represent in the legend for every figure.

This information was added to the figure legends.

The sequence maps of the constructs developed in the study need to be well annotated with major features clearly labeled before the final submission.

We improved annotation of the DNA constructs and now also provide a geneious construct library in addition to the annotated genbank files.

Reviewers' Comments:

Reviewer #1:

Remarks to the Author:

Thanks for addressing my comments and adding the additional supp. figure. The issues have all been addressed satisfactorily. I note that the figure legend of S6 appears to have a typo "now".

Keep up the fascinating science!

Reviewer #2:

Remarks to the Author:

All of my comments have been addressed.

REVIEWERS' COMMENTS

Reviewer #1 (Remarks to the Author):

Thanks for addressing my comments and adding the additional supp. figure. The issues have all been addressed satisfactorily. I note that the figure legend of S6 appears to have a typo "now".

We corrected this in the text.

Keep up the fascinating science!

We thank the reviewer for their positive comment.

Reviewer #2 (Remarks to the Author):

All of my comments have been addressed.

We thank the reviewer for their positive comment.